# Supervised autoencoders: Improving generalization performance with unsupervised regularizers

**Lei Le**
Department of Computer Science
Indiana University
Bloomington, IN
leile@iu.edu

**Andrew Patterson and Martha White**
Department of Computing Science
University of Alberta
Edmonton, AB T6G 2E8, Canada
{ap3, whitem}@ualberta.ca

## Abstract

Generalization performance is a central goal in machine learning, with explicit generalization strategies needed when training over-parametrized models, like large neural networks. There is growing interest in using multiple, potentially auxiliary tasks, as one strategy towards this goal. In this work, we theoretically and empirically analyze one such model, called a supervised auto-encoder: a neural network that jointly predicts targets and inputs (reconstruction). We provide a novel generalization result for linear auto-encoders, proving uniform stability based on the inclusion of the reconstruction error—particularly as an improvement on simplistic regularization such as norms. We then demonstrate empirically that, across an array of architectures with a different number of hidden units and activation functions, the supervised auto-encoder compared to the corresponding standard neural network never harms performance and can improve generalization.

## 1 Introduction

Generalization is a central concept in machine learning: learning functions from a finite set of data, that can perform well on new data. Generalization bounds have been characterized for many functions, including linear functions [1], and those with low-dimensionality [2, 3] and functions from reproducing kernel Hilbert spaces [4]. Many of these bounds are obtained through some form of regularization, typically $\ell_2$ regularization [5, 6] or from restricting the complexity of the function class such as by constraining the number of parameters [1].

Understanding generalization performance is particularly critical for powerful function classes, such as neural networks. Neural networks have well-known overfitting issues, with common strategies to reduce overfitting including drop-out [7–9], early stopping [10] and data augmentation [11, 12], including adversarial training [13] and label smoothing [14]. Many layer-wise regularization strategies have also been suggested for neural networks, such as with layer-wise training [15, 16], pre-training with layer-wise additions of either unsupervised learning or supervised learning [15] and the use of auxiliary variables for hidden layers [17].

An alternative direction that has begun to be explored is to instead consider regularization with the addition of tasks. Multi-task learning [18] has been shown to improve generalization performance, from early work showing learning tasks jointly reduces the required number of samples [19, 20] and later work particularly focused on trace-norm regularization on the weights of a linear, single hidden-layer neural network for a set of tasks [21–23]. Some theoretical work has also been done for auxiliary tasks [24], with the focus of showing that the addition of auxiliary tasks can improve the representation and so generalization. In parallel, a variety of experiments have demonstrated the utility of adding layer-wise unsupervised errors as auxiliary tasks [15, 16, 25–27]. Auxiliary tasks have also been explored through the use of hints for neural networks [28, 18].

In this work, we investigate an auxiliary-task model for which we can make generalization guarantees, called a *supervised auto-encoder* (SAE). A SAE is a neural network that predicts both inputs and outputs, and has been previously shown empirically to provide significant improvements when used in a semi-supervised setting [16] and deep neural networks [29]. We provide a novel uniform stability result, showing that linear SAE—which consists of the addition of reconstruction error to a linear neural network— provides uniform stability and so a bound on generalization error. We show that the stability coefficient decays similarly to the stability coefficient under $\ell_2$ regularization [5], providing effective generalization performance but avoiding the negative bias from shrinking coefficients. The reconstruction error may incur some bias, but is related to the prediction task and so is more likely to prefer a more robust model amongst a set of similarly effective models for prediction. This bound, to the best of our knowledge, is (a) one of the first bounds demonstrating that supervised dimensionality reduction architectures can provide improved generalization performance and (b) provides a much tighter bound than is possible from applying generalization results from multi-task learning [21–23] and learning with auxiliary tasks [24]. Finally, we demonstrate empirically that adding reconstruction error never harms performance compared to the corresponding neural network model, and in some cases can significantly improve classification accuracy.

## 2 Supervised autoencoders and representation learning

We consider a supervised learning setting, where the goal is to learn a function for a vector of inputs $\mathbf{x} \in \mathbb{R}^d$ to predict a vector of targets $\mathbf{y} \in \mathbb{R}^m$. The function is trained on a finite batch of i.i.d. data, $(\mathbf{x}_1, \mathbf{y}_1), \ldots, (\mathbf{x}_t, \mathbf{y}_t)$, with the aim to predict well on new samples generated from the same distribution. To do well in prediction, a common goal is *representation learning*, where the input $\mathbf{x}_i$ are first transformed into a new representation, for which it is straightforward to learn a simple predictor—such as a linear predictor.

Auto-encoders (AE) are one strategy to extract a representation. An AE is a neural network, where the outputs are set to $\mathbf{x}$, the inputs. By learning to reconstruct the input, the AE extracts underlying or abstract attributes that facilitate accurate prediction of the inputs. Linear auto-encoders with a single hidden layer are equivalent to principle components analysis [30][31, Theorem 12.1], which finds (orthogonal) explanatory factors for the data. More generally, nonlinear auto-encoders have indeed been found to extract key attributes, including high-level features [32] and Gabor-filter features [33].

A supervised auto-encoder (SAE) is an auto-encoder with the addition of a supervised loss on the representation layer. For a single hidden layer, this simply means that a supervised loss is added to the output layer, as in Figure 1. For a deeper auto-encoder, the innermost (smallest)[1] layer would have a supervised loss added to it—the layer that would usually be handed off to the supervised learner after training the AE. More formally, consider a linear SAE, with a single hidden layer of size $k$. The weights for the first layer are $\mathbf{F} \in \mathbb{R}^{d \times k}$. The weight for the output layer consist of weights $\mathbf{W}_p \in \mathbb{R}^{k \times m}$ to predict $\mathbf{y}$ and $\mathbf{W}_r \in \mathbb{R}^{k \times d}$ to reconstruct $\mathbf{x}$. Let $L_p$ be the supervised (primary) loss and $L_r$ the loss for the reconstruction error. For example, in regression, both losses might be the squared error, resulting in the objective

$$\frac{1}{t} \sum_{i=1}^{t} \left[ L_p(\mathbf{W}_p \mathbf{F} \mathbf{x}_i, \mathbf{y}_i) + L_r(\mathbf{W}_r \mathbf{F} \mathbf{x}_i, \mathbf{x}_i) \right] = \frac{1}{2t} \sum_{i=1}^{t} \left[ \| \mathbf{W}_p \mathbf{F} \mathbf{x}_i - \mathbf{y}_i \|_2^2 + \| \mathbf{W}_r \mathbf{F} \mathbf{x}_i - \mathbf{x}_i \|_2^2 \right]. \quad (1)$$

The addition of a supervised loss to the auto-encoder should better direct representation learning towards representations that are effective for the desired tasks. Conversely, solely training a representation according to the supervised tasks, like learning hidden layers in an neural network, is likely an under-constrained problem, and will find solutions that can well fit the data but that do not find underlying patterns in the data and do not generalize well. In this way, the combination of the two losses has the promise to both balance extracting underlying structure, as well as providing accurate prediction performance. There have been several empirical papers that have demonstrated the capabilities of semi-supervised autoencoders [16, 27, 34]. Those results focus on the semi-supervised component, where the use of auto-encoders enables the representation to be trained with more unlabeled data. In this paper, however, we would like to determine if even in the purely supervised setting, the addition of reconstruction error can have a benefit for generalization.

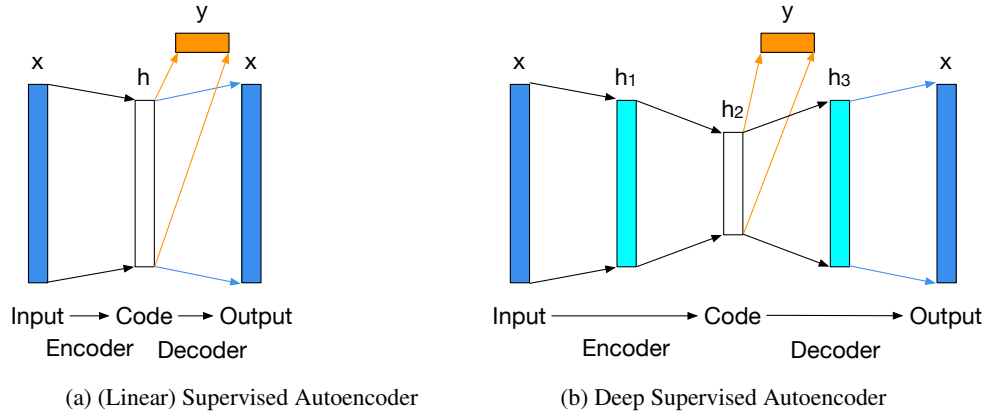

(a) (Linear) Supervised Autoencoder&emsp;&emsp;&emsp;(b) Deep Supervised Autoencoder

Figure 1: Two examples of Supervised Autoencoders, and where the supervised component—the targets $y$—are included. We provide generalization performance results for linear SAEs, represented by (a) assuming a linear activation to produce the hidden layer, with arbitrary convex losses on the output layer, such as the cross-entropy for classification. We investigate more general architectures in the experiments, including single-hidden layer SAEs, represented by (a) with nonlinear activations to produce the hidden layer, and deep SAEs, depicted in (b).

## 3&emsp;Uniform stability and generalization bounds for SAE

In this section, we show that including the reconstruction error theoretically improves generalization performance. We show that linear supervised auto-encoders are uniformly stable, which means that there is a small difference between models learned for any subsample of data, which differ in only one instance. Uniformly stable algorithms are known to have good generalization performance [5]. Before showing this result, we discuss a few alternatives to justify why we pursue uniform stability.

There are at least two alternative strategies that could be considered to theoretically analyze these models: using a multi-task analysis and characterizing the Rademacher complexity of the supervised auto-encoder function class. The reconstruction error can in-fact be considered as multiple tasks, where the multiple tasks help regularize or constrain the solution [35]. Previous results for multi-task learning [21–23] demonstrate improved generalization error bounds when learning multiple tasks jointly. Unfortunately, these bounds show performance is improved *on average* across tasks. For our setting, we only care about the primary tasks, with the reconstruction error simply included as an auxiliary task to regularize the solution. An average improvement might actually mean that performance on the primary task degrades with inclusion of these other tasks. Earlier multi-task work did consider improvement for each task [36], but assumed different randomly generated features for each task and all tasks binary classification problem, which does not match this setting.

Another strategy is to characterize Rademacher complexity of supervised auto-encoders. There has been some work characterizing the Rademacher complexity of unsupervised dimensionality reduction techniques [3, Theorem 3.1]. To the best of our knowledge, however, there as yet does not appear to be an analysis on complexity of supervised dimensionality reduction techniques. There is some work on supervised dimension reduction [2, 3]; however, this analysis assumes a dimensionality reduction step followed by a supervised learning step, rather than a joint training procedure.

For these reasons, we pursue a third direction, where we treat the reconstruction error as a regularizer to promote stability. Uniform stability has mainly been obtained using norm-based regularization strategies, such as $\ell_2$. More recently, Liu et al. [24] showed that auxiliary tasks—acting as regularizers—could also provide uniform stability. Because reconstruction error can be considered to be an auxiliary task, our analysis resembles this auxiliary-task analysis. However, there are key differences, as the result by Liu et al. [24] would be uninteresting if simply applied directly to our setting. In particular, the uniform stability bound would not decay with the number of samples. The bound decays proportionally to the number of samples for the primary task, but in the numerator contains the maximum number of samples for an auxiliary task. For us, this maximum number is exactly the same number of samples as for the primary task, and so they would cancel, making the bound independent of the number of samples.

**Primary Result**

We now show that the parameter shared by the primary task and reconstruction error—the forward model $\mathbf{F}$—does not change significantly with the change of one sample. This shows that linear SAEs have uniform stability, which then immediately provides a generalization bound from [5, Theorem 12]. The proofs are provided in the appendix, for space.

Let $L_p$ corresponds to the primary part (supervised part) of the loss, with weights $\mathbf{W}_p$, and $L_r$ correspond to the auxiliary tasks that act as regularizers (the reconstruction error), with weights $\mathbf{W}_r$. The full loss can be written

$$L(\mathbf{F}) = \frac{1}{t} \sum_{i=1}^{t} L_p\left(\mathbf{W}_p \mathbf{F} \mathbf{x}_i, \mathbf{y}_{p,i}\right) + L_r\left(\mathbf{W}_r \mathbf{F} \mathbf{x}_i, \mathbf{y}_{r,i}\right). \tag{2}$$

For our specific setting, $\mathbf{y}_r = \mathbf{x}$. We use more general notation, however, both to clarify the difference between the inputs and outputs, and for future extensions to this theory for other (auxiliary) targets $\mathbf{y}_r$. The loss where the $m$-th sample $(\mathbf{x}_m, \mathbf{y}_m)$ is replaced by a random new instance $(\mathbf{x}'_m, \mathbf{y}'_m)$ is

$$L_m(\mathbf{F}){=}\tfrac{1}{t}\Big[L_p\big(\mathbf{W}_p \mathbf{F} \mathbf{x}'_m, \mathbf{y}'_{p,m}\big){+}L_r\big(\mathbf{W}_r \mathbf{F} \mathbf{x}'_m, \mathbf{y}'_{r,m}\big){+}\sum_{i=1,i\neq m}^{t} L_p(\mathbf{W}_p \mathbf{F} \mathbf{x}_i, \mathbf{y}_{p,i}){+}L_r(\mathbf{W}_r \mathbf{F} \mathbf{x}_i, \mathbf{y}_{r,i})\Big].$$

If we let $\mathbf{F}, \mathbf{F}_m$ correspond to the optimal forward models for these two losses respectively, then the algorithm is said to be $\beta$-uniformly stable if the difference in loss value for these two models for any point $(\mathbf{x}, \mathbf{y})$ is bounded by $\beta$ with high-probability

$$|L_p\left(\mathbf{W}_p \mathbf{F}_m \mathbf{x}, \mathbf{y}_p\right) - L_p\left(\mathbf{W}_p \mathbf{F} \mathbf{x}, \mathbf{y}_p\right)| \leq \beta$$

To obtain uniform stability, we will need to make several assumptions. The first common assumption is to assume bounded spaces, for the data and learned variables.

**Assumption 1.** *The features satisfy $\|\mathbf{x}\|_2 \leq B_\mathbf{x}$ and the primary targets satisfy $\|\mathbf{y}_p\|_2 \leq B_\mathbf{y}$. The parameters spaces are bounded,*

$$\mathcal{W} = \{(\mathbf{W}_p, \mathbf{W}_r) \in \mathbb{R}^{k \times m} : \|\mathbf{W}_p\|_F \leq B_{\mathbf{W}_p}, \|\mathbf{W}_r\|_F \leq B_{\mathbf{W}_r}\}$$
$$\mathcal{F} = \{\mathbf{F} \in \mathbb{R}^{d \times k} : \|\mathbf{F}\|_F \leq B_\mathbf{F}\}$$

*for some positive constants $B_\mathbf{x}, B_\mathbf{y}, B_\mathbf{F}, B_{\mathbf{W}_p}, B_{\mathbf{W}_r}$, where $\|\cdot\|_F$ denotes Frobenius norm, namely the square root of the sum of the squares of all elements.*

For SAE, $\mathbf{y}_r = \mathbf{x}$, and so $\|\mathbf{x}\|_2 \leq B_\mathbf{x}$ implies that $\|\mathbf{y}_r\|_2 \leq B_\mathbf{x}$.

Second, we need to ensure that the reconstruction error is both strongly convex and Lipschitz. The next two assumptions are satisfied, for example, by the $\ell_2$ loss, $L_r(\hat{\mathbf{y}}, \mathbf{y}) = \|\hat{\mathbf{y}} - \mathbf{y}\|_2^2$.

**Assumption 2.** *The reconstruction loss $L_r(\cdot, \mathbf{y})$ is $\sigma_r$-admissible, i.e., for possible predictions $\hat{\mathbf{y}}, \hat{\mathbf{y}}'$*

$$|L_r\left(\hat{\mathbf{y}}, \mathbf{y}\right) - L_r\left(\hat{\mathbf{y}}', \mathbf{y}\right)| \leq \sigma_r \|\hat{\mathbf{y}} - \hat{\mathbf{y}}'\|_2.$$

**Assumption 3.** $L_r(\cdot, \mathbf{y})$ *is $c$-strongly-convex* $\langle \hat{\mathbf{y}} - \hat{\mathbf{y}}', \nabla L_r\left(\hat{\mathbf{y}}, \mathbf{y}\right) - \nabla L_r\left(\hat{\mathbf{y}}', \mathbf{y}\right)\rangle \geq c\|\hat{\mathbf{y}} - \hat{\mathbf{y}}'\|_2^2$

The growth of the primary loss also needs to be bounded; however, we can use a less stringent requirement than admissibility.

**Assumption 4.** *For some $\sigma_p > 0$, for any $\mathbf{F}, \mathbf{F}_m \in \mathcal{F}$,*

$$|L_p\left(\mathbf{W}_p \mathbf{F}_m \mathbf{x}, \mathbf{y}_p\right) - L_p\left(\mathbf{W}_p \mathbf{F} \mathbf{x}, \mathbf{y}_p\right)| \leq \sigma_p \|\mathbf{W}_r(\mathbf{F}_m - \mathbf{F})\mathbf{x}\|_2$$

This requirement should be less stringent because we expect generally that for two forward models $\mathbf{F}, \mathbf{F}_m, \|\mathbf{W}_p(\mathbf{F} - \mathbf{F}_m)\mathbf{x}\|_2 \leq \|\mathbf{W}_r(\mathbf{F} - \mathbf{F}_m)\mathbf{x}\|_2$. The matrix $\mathbf{W}_p \in \mathbb{R}^{m \times k}$ projects the vector $\mathbf{d} = (\mathbf{F} - \mathbf{F}_m)\mathbf{x}$ into a lower-dimensional space, whereas $\mathbf{W}_r \in \mathbb{R}^{d \times k}$ projects $\mathbf{d}$ into a higher-dimensional space. Because the nullspace of $\mathbf{W}_p$ is likely larger, it is more likely that $\mathbf{W}_p$ will send a non-zero $\mathbf{d}$. In fact, if $\mathbf{W}_r$ is full rank—which occurs if $k$ is less than or equal to the intrinsic rank of the data—then we can guarantee this assumption for some $\sigma_p$ as long as $L_p$ is $\sigma$-admissible, where likely $\sigma_p$ can be smaller than $\sigma$. In Corollary 1, we specify the value of $\sigma_p$ under a full rank $\mathbf{W}_r$ and $\sigma$-admissible $L_p$.

Finally, we assume that there is a representative set of feature vectors in the sampled data, both in terms of feature vectors (Assumption 5) as well as loss values (Assumption 6).

**Assumption 5.** *There exists a subset*

$$B = \{\mathbf{b}_1, \mathbf{b}_2, ..., \mathbf{b}_n\} \subset \{\mathbf{x}_1, \mathbf{x}_2, ..., \mathbf{x}_t\}$$

*such that with high probability any sampled feature vector $\mathbf{x}$ can be reconstructed by $B$ with a small error: $\mathbf{x} = \sum_{i=1}^{n} \alpha_i \mathbf{b}_i + \eta$ where $\alpha_i \in \mathbb{R}, \sum_{i=1}^{n} \alpha_i^2 \leq r, \|\eta\|_2 \leq \frac{\epsilon}{t}$.*

Assumption 5 is similar to [24, Assumption 1], except for our setting the features are the same for all the tasks and the upper bound of $\|\eta\|$ decreases as $\frac{1}{t}$. This is a reasonable assumption since more samples in the training set make it more likely to be able to reconstruct any $\mathbf{x}$ that will be observed with non-negligible probability. In many cases, $\eta = 0$ is a mild assumption, as once $d$ independent vectors $\mathbf{b}_i$ are observed, $\eta = 0$.

This representative set of points also needs to be representative in terms of the reconstruction error. In particular, we need the average reconstruction error of the representative points to be upper bounded by some constant factor of the average reconstruction error under the training set.

**Assumption 6.** *For any two datasets $S, S_m$, where $S_m$ has the $m$-th sample replaced with a random new instance, let $\mathbf{F}, \mathbf{F}_m$ be the corresponding optimal forward models. Let $N$ contain only the reconstruction errors, without the sample that is replaced*

$$N(\mathbf{F}) = \frac{1}{t} \sum_{i=1, i \neq m}^{t} L_r\left(\mathbf{W}_r \mathbf{F} \mathbf{x}_i, \mathbf{y}_{r,i}\right) \tag{3}$$

*and $N_b$ be the reconstruction error for the representative points*

$$N_b(\mathbf{F}) = \frac{1}{n} \sum_{i=1}^{n} L_r\left(\mathbf{W}_r \mathbf{F} \mathbf{b}_i, \mathbf{y}_{r,b_i}\right) \tag{4}$$

*where $\mathbf{y}_{r,b_i}$ is the reconstruction target for representative point $\mathbf{b}_i$. Then, there exists $a > 0$ such that for any small $\alpha > 0$,*

$$[N_b(\mathbf{F}) - N_b((1-\alpha)\mathbf{F} + \alpha\mathbf{F}_m)] + [N_b(\mathbf{F}_m) - N_b((1-\alpha)\mathbf{F}_m + \alpha\mathbf{F})]$$
$$\leq a\left[N(\mathbf{F}) - N((1-\alpha)\mathbf{F} + \alpha\mathbf{F}_m)\right] + a\left[N(\mathbf{F}_m) - N((1-\alpha)\mathbf{F}_m + \alpha\mathbf{F})\right].$$

The above assumption does not require that the difference under $N$ and $N_b$ be small for the two $\mathbf{F}$ and $\mathbf{F}_m$; rather, it only requires that the increase or decrease in error at the two points $\mathbf{F}_m$ and $\mathbf{F}$ are similar for $N$ and $N_b$. Both the right-hand-side and left-hand-side in the assumption are nonnegative, because of the convexity of $N$ and $N_b$. Even if $N$ is higher at $\mathbf{F}$ than $\mathbf{F}_m$, and $N_b$ is the opposite, the above bound can hold, because it simply requires that the difference of $N_b$ between $\mathbf{F}_m$ and $\mathbf{F}$ be bounded above by the difference of $N$ between $\mathbf{F}$ and $\mathbf{F}_m$, up to some constant factor $a$. This assumption is key, because we will need to use $N_b$ to ensure that the bound decays with $t$, where $N_b$ is only dependent on the number of representative points, unlike $N$.

We can now provide the key result: SAE has uniform stability wrt the shared parameters $\mathbf{F}$.

**Theorem 1.** *Under Assumptions 1-6, for a randomly sampled $\mathbf{x}, \mathbf{y}$, with high probability*

$$|L_p(\mathbf{W}_p \mathbf{F}_m \mathbf{x}, \mathbf{y}) - L_p(\mathbf{W}_p \mathbf{F} \mathbf{x}, \mathbf{y})| \leq \frac{a(\sigma_r + \sigma_p)n\sigma_p}{ct}\left(r + \sqrt{r^2 + \frac{4\epsilon c B_{\mathbf{W}_r} B_{\mathbf{F}} r}{a(\sigma_r + \sigma_p)n}}\right) + \frac{2\epsilon \sigma_p B_{\mathbf{W}_r} B_{\mathbf{F}}}{t} \tag{5}$$

**Remark:** We similarly get $O(\frac{1}{t})$ upper bound on instability from Bousquet and Elisseeff [5], but without requiring the $\ell_2$ regularizer. The $\ell_2$ indiscriminately reduces the magnitude of the weights; the reconstruction error, on the other hand, regularizes, but potentially without strongly biasing the solution. It can select amongst a set of possible forward models that predict the targets almost equally well, but that also satisfy reconstruction error. A hidden representation that is useful for reconstructing the inputs is likely to also be effective for predicting the targets—which are a function of the inputs.

**Corollary 1.** *In Assumption 4, if $\mathbf{W}_p \in \mathbb{R}^{m \times k}, \mathbf{W}_r \in \mathbb{R}^{d \times k}, d \geq k \geq m, \mathbf{W}_r$ is full rank, $L_p$ is $\sigma$-admissible, then for $\mathbf{W}_r^{-1}$ the inverse matrix of the first $k$ rows of $\mathbf{W}_r$, $\sigma_p = \sigma \|\mathbf{W}_p\|_F \|\mathbf{W}_r^{-1}\|_F$.*

Finally, we provide a few specific bounds, for particular $L_r$ and $L_p$, to show how this more general bound can be used (shown explicitly in Appendix B). For example, for a least-squares reconstruction loss $L_r$, $c = 2$ and $\sigma_r = 2B_{\mathbf{W}_r} B_{\mathbf{F}} B_{\mathbf{x}} + 2B_{\mathbf{x}}$.

# 4   Experiments with SAE: Utility of reconstruction error

We now empirically test the utility of incorporating the reconstruction error into NNs, as a method for regularization to improve generalization performance. Our goal is to investigate the impact of the reconstruction error, and so we use the same architecture for SAE and NN, where the only difference is the use of reconstruction error. We test several different architectures, namely single-hidden layer SAEs with different activations, adding non-linearity with kernels before using a linear SAE and a deep SAE with a bottleneck, namely a hidden layer with smaller size than that of the previous layer.

**Experimental setup and Datasets.**  We used 10-fold cross-validation to choose the best meta-parameters for each algorithm on each dataset. The meta-parameters providing the highest classification accuracy averaged across folds are chosen. Using the meta-parameters chosen by cross-validation, we report the average accuracy and standard error across 20 runs, each with a different randomly sampled training-testing splits. A new training-testing split is generated by shuffling all data points together and selecting the first samples to be the training set, and the remaining to be the testing set.

**SUSY** is a high-energy particle physics dataset [37]. The goal is to classify between a process where supersymmetric particles are produced, and a background process where no detectable particles are produced. SUSY was generated to discover hidden representations of raw sensor features for classification [37], and has 8 features and 5 million data points.

**Deterding** is a vowel dataset [38] containing 11 steady-state vowels of British English spoken by 15 speakers. Every speaker pronounced each of the eleven vowel sounds six times giving 990 labeled data points. The goal is to classify the vowel sound for each spoken vowel, where each speech signal is converted into a 10-dimensional feature vector using log area ratios based on linear prediction coefficients. We normalized each feature between 0 and 1 through Min-Max scaling.

**CIFAR**-10 is an image dataset [39] with 10 classes and 60000 32x32 color images. The classes include objects like horses, deer, trucks and airplanes. For each of the training-test splits, we used a random subset of 50,000 images for training and 10,000 images for testing. We preprocessed the data by averaging together the three colour channels creating gray-scale images to speed up computation.

**MNIST** is a dataset [40] of 70000 examples of 28x28 images of handwritten digits from 0 to 9.

We would like to note that for these two benchmark datasets—CIFAR and MNIST—impressive performance has been achieved, such as with a highly complex, deep neural network model for CIFAR [41]. Here, however, we use these datasets to investigate a variety of models, rather than to match performance of the current state-of-the-art. We do not use the provided single training-testing split, but rather treat these large datasets as an opportunity to generate many (different) training-test splits for a thorough empirical investigation.

**Overall results.** Figure 2 shows the performance of SAE versus NN. On the Deterding, SUSY and MNIST datasets, we compare them in three different architectures. First, we compare linear SAE with linear NN, where there is no activation function from the input to the hidden layer. Second, we nonlinearly transform the data with radial basis functions—a Gaussian kernel—and then use linear SAE and linear NNs. The kernel expansion enables nonlinear functions to be learned, despite the fact that the learning step can still benefit from the optimality results provided for linear SAE. Third, we use nonlinear activation functions, sigmoid and ReLu, from the input to the hidden layer. Though this is outside the scope of the theoretical characterization, it is a relatively small departure and important to understand the benefits of the reconstruction error for at least simple nonlinear networks. We investigate only networks with single hidden layers as a first step, and to better match the networks characterized in the theoretical guarantees.

Overall, we find that SAE improves performance across settings, in some cases by several percent. Getting even an additional 1% in classification accuracy with just the addition of reconstruction error to relatively simple models is a notable result. We summarize these results in Figure 2 and Table 1. SAE and NN with the same architecture have similar sample variances, so we use a t-test for statistical significance. For all pairs but one, the average accuracy of SAE is statistically significantly higher than that of NN, with significance level 0.0005, though in some cases the differences are quite small, particularly on SUSY and MNIST. In other cases, particularly in kernel representations in Deterding, SAE significantly outperformed NN, with a jump by 18% in classification accuracy. Because we attempted to standardize the models, differing only in SAE using reconstruction error, these results indicate that the reconstruction error has a clear positive impact on generalization performance.

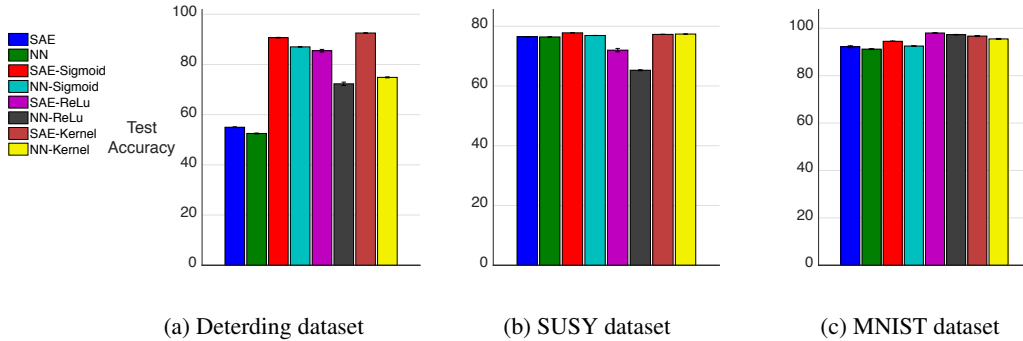

|   | (a) Deterding dataset | (b) SUSY dataset | (c) MNIST dataset |
|---|---|---|---|

Figure 2: Test accuracy of a three layer neural network (NN) and our supervised auto-encoder model (SAE), on three datasets. We focus on the impact of using reconstruction error, and compare SAE and NN with a variety of nonlinear structures, including sigmoid (SAE-Sigmoid and NN-Sigmoid), ReLu (SAE-ReLu and NN-ReLu) and Gaussian kernel (SAE-Kernel and NN-Kernel). Though not showing the results in the figure, we also tried initializing NN with pre-trained autoencoders and the performance is similar to NN, thus outperformed by SAE as well. Overall, SAE consistently outpeforms NNs, though in some cases the advantage is small. Details are shown in Table 1.

|  | Deterding | | SUSY | | MNIST | |
|---|---|---|---|---|---|---|
|  | Average Accuracy $\pm$ Standard Error | | Average Accuracy $\pm$ Standard Error | | Average Accuracy $\pm$ Standard Error | |
|  | Test | Training | Test | Training | Test | Training |
| SAE | **54.98 $\pm$ 0.18** | **63.34 $\pm$ 0.17** | **76.48 $\pm$ 0.01** | **76.50 $\pm$ 0.03** | **92.20 $\pm$ 0.40** | **93.70 $\pm$ 0.30** |
| NN | 52.50 $\pm$ 0.17 | 61.05 $\pm$ 0.14 | 76.41 $\pm$ 0.02 | 76.42 $\pm$ 0.02 | 91.20 $\pm$ 0.20 | 92.50 $\pm$ 0.22 |
| SAE-Sigmoid | **90.67 $\pm$ 0.12** | **99.38 $\pm$ 0.03** | **77.79 $\pm$ 0.02** | **77.80 $\pm$ 0.01** | **94.50 $\pm$ 0.10** | **96.35 $\pm$ 0.05** |
| NN-Sigmoid | 87.00 $\pm$ 0.14 | 97.62 $\pm$ 0.05 | 76.90 $\pm$ 0.03 | 76.90 $\pm$ 0.01 | 92.50 $\pm$ 0.10 | 96.20 $\pm$ 0.04 |
| SAE-ReLu | **85.47 $\pm$ 0.52** | **90.22 $\pm$ 0.41** | **71.99 $\pm$ 0.58** | 72.04 $\pm$ 0.33 | **98.00 $\pm$ 0.10** | **98.25 $\pm$ 0.08** |
| NN-ReLu | 72.29 $\pm$ 0.67 | 78.76 $\pm$ 0.08 | 65.27 $\pm$ 0.17 | **75.03 $\pm$ 0.11** | 97.30 $\pm$ 0.10 | 98.10 $\pm$ 0.09 |
| SAE-Kernel | **92.52 $\pm$ 0.10** | **93.15 $\pm$ 0.11** | 77.27 $\pm$ 0.06 | 77.31 $\pm$ 0.12 | **96.70 $\pm$ 0.20** | **97.40 $\pm$ 0.18** |
| NN-Kernel | 74.85 $\pm$ 0.20 | 82.37 $\pm$ 0.41 | **77.38 $\pm$ 0.06** | **77.42 $\pm$ 0.06** | 95.50 $\pm$ 0.20 | 96.20 $\pm$ 0.20 |

Table 1: The percentage accuracy for the results presented in Figure 2. SAE outperforms NNs in terms of average test accuracy across settings. The only exception is the Gaussian kernel on SUSY, where the advantage of NN-Kernel is extremely small. We report train accuracies for further insights and completeness. Note that though there is some amount of overfitting occurring, the models were given the opportunity to select a variety of regularization parameters for $\ell_2$ regularization as well as dropout using cross-validation.

In the next few sections, we highlight certain properties of interest, in addition to these more general performance results. We highlight robustness to overfitting as model complexity is increased, for both nonlinear activations and kernel transformations. For these experiments, we choose CIFAR, since it is a more complex prediction problem with a large amount of data. We then report preliminary conclusions on the strategy of over-parametrizing and regularizing, rather than using bottleneck layers. Finally, we demonstrate the structure extracted by SAE, to gain some insight into the representation.

**Robustness to overfitting.** We investigate the impact of increasing the hidden dimension on CIFAR, with sigmoid and ReLu activation functions from the input to the hidden layer. The results are summarized in Figures 3a and 3b, where the hidden dimension is increased from 20 to as large as 10 thousand. Both results indicate that SAE can better take advantage of increasing model complexity, where (a) the NN clearly overfit and obtained poor accuracy with a sigmoid transfer and (b) SAE gained a 2% accuracy improvement over NNs when both used a ReLu transfer.

**Results with kernels.** The overall conclusion is that SAE can benefit much more from model complexity given by kernel representations, than NNs. In Table 1, the most striking difference between SAE and NNs with kernels occurs for the Deterding dataset. SAE outperforms NN by an entire 18%, going from 75% test accuracy to 92% test accuracy. For SUSY, SAE and NNs were essentially tied; but for that dataset, all the nonlinear architectures performed very similarly, suggesting little improvement could be gained.

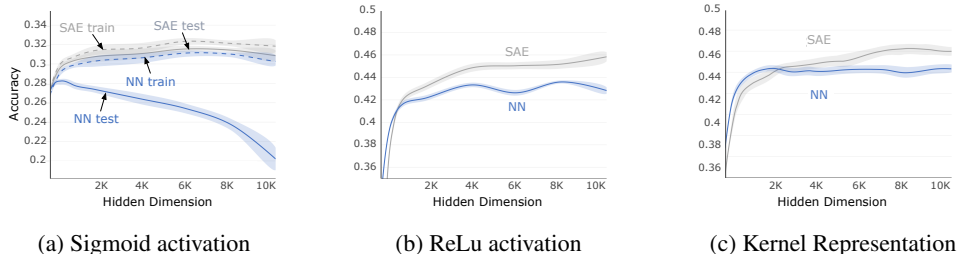

|  (a) Sigmoid activation | (b) ReLu activation | (c) Kernel Representation |

Figure 3: Test accuracy of SAE and NN with a variety of nonlinear architectures on CIFAR, with increasing model complexity. For the sigmoid and relu, the hidden dimension is increased; for kernels, the number of centers is increased. **(a)** For the sigmoid activation, the NN suffers noticeably from overfitting as the hidden dimension increases, whereas SAE is robust to the increase in model complexity. **(b)** For the ReLu activation, under low model complexity, SAE performed more poorly than the NN. However, given a larger hidden dimension—about a half as large as the input dimension— it reaches the same level of performance and then is better able to take advantage of the increase model complexity. The difference of about 2% accuracy improvement for such a simple addition—the reconstruction error—is a striking result. **(c)** The result here is similar to ReLu. Note that the size of the hidden dimension corresponds to 10% of the number of centers.

On CIFAR, we also investigated the impact of increasing the number of kernel centers, which correspondingly increases model parameters and model complexity. We fixed the hidden dimension to 10% of the number of centers, to see if the SAE could still learn an appropriate model even with an aggressive bottleneck, namely a hidden layer with a relatively very small size, making it hard to reduce the reconstruction error. This helps to verify the hypothesis that the reconstruction error does not incur much bias as a regularizer, and test a more practical setting where an aggressive bottleneck can significantly speed up computation and convergence rate. For the NN, because the number of targets is 10, once the hidden dimension $k \geq 10$, the bottleneck should have little to no impact on performance, which is what we observe. The result is summarized in Figure 3c, which shows that SAE initially suffers when model complexity is low, but then surpasses the NN with increasing model complexity. In general, we anticipate the effects with kernels and SAE to be more pronounced with more powerful selection of kernels and centers.

**Demonstration of SAE with a Deep Architecture.** We investigate the effects of adding the reconstruction loss to deep convolutional models on CIFAR. We use a network with two convolutional layers of sizes $\{32, 64\}$ and 4 dense layers of sizes $\{2048, 512, 128, 32\}$ with ReLu activation. Unlike our previous experiments we do not use grey-scale CIFAR, but instead use all three color channels for the deep networks to make maximal use of the convolutional layers.

As shown in Figure 4, SAE outperforms NN consistently in both train and test accuracies, suggesting that SAE is able to find a different and better solution than NN in the optimization on the training data and generalize well on the testing data. We show the performance of SAE with decreasing weight on the predictive loss, which increases the effect of the reconstruction error. Interestingly, a value of 0.01 performs the best, but began to degrade with lower values. At the extreme, for a weight of 0 which corresponds to an Autoencoder, performance is significantly worse, so the combination of both is necessary. We discuss other variants we tried in the caption for Figure 4, but the conclusions remain consistent: SAE improves generalization performance over NNs.

## 5   Conclusion

In this paper, we systematically investigated supervised auto-encoders (SAEs), as an approach to using unsupervised auxiliary tasks to improve generalization performance. We showed theoretically that the addition of reconstruction error improves generalization performance, for linear SAEs. We showed empirically, across four different datasets, with a variety of architectures, that SAE never harms performance but in some cases can significantly improve performance, particularly when using kernels and under ReLu activations, for both shallow and deep architectures.

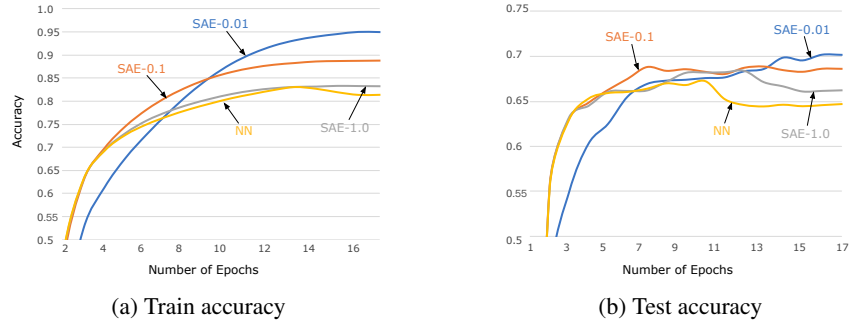

(a) Train accuracy            (b) Test accuracy

Figure 4: Train and Test accuracy of SAE and NN with a deep architecture. The numbers 0.01, 0.1 and 1.0 denote the weights on the prediction error, with a constant weights of 1.0 on the reconstruction error. We also compared to Auto-encoders, with a two-stage training strategy where the auto-encoder is trained first, with the representation then used for the supervised learner, but this performed poorly (about 0.4 testing accuracy). We additionally investigated both dropout and $\ell_2$ regularization. We find that dropout increases the variance of independent runs, and improves each algorithm by approximately three percentage points over its reported test set accuracy. Using $\ell_2$ regularization did not improve performance. Under both dropout and $\ell_2$, the advantage of SAE over NN in both train and test accuracies remained consistent, and so these graphs are representative for those additional settings. Finally, we additionally compared to the ResNet-18 architecture [42]. For a fair comparison, we do not use the image augmentation originally used in training ResNet-18. We find that ResNet-18, with nearly double the total learnable parameters, achieved only two percentage points higher on the test set accuracy than our SAE with reconstructive loss.

## Footnotes

[1]The size of the learned representations for deep, nonlinear AEs does not have to be small, but it is common to learn such a lower-dimensional representations. For linear SAEs, the hidden layer size $k < d$, as otherwise trivial solutions like the replication of the input are able to minimize the reconstruction error.

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
