[Supplementary Material · supplementary.pdf]

# A   Proof of Main Theorem

**Theorem 1** *Under Assumptions 1-6, for a randomly sampled* $\mathbf{x}, \mathbf{y}$, *with high probability*

$$|L_p(\mathbf{W}_p\mathbf{F}_m\mathbf{x}, \mathbf{y}) - L_p\left(\mathbf{W}_p\mathbf{F}\mathbf{x}, \mathbf{y}\right)| \leq \frac{a(\sigma_r+\sigma_p)n\sigma_p}{ct}\left(r + \sqrt{r^2 + \frac{4\epsilon cB_{\mathbf{W}_r}B_{\mathbf{F}}r}{a(\sigma_r+\sigma_p)n}}\right) + \frac{2\epsilon\sigma_pB_{\mathbf{W}_r}B_{\mathbf{F}}}{t} \quad (6)$$

**Proof:** From Assumption 4,

$$|L_p\left(\mathbf{W}_p\mathbf{F}_m\mathbf{x}, \mathbf{y}_p\right) - L_p\left(\mathbf{W}_p\mathbf{F}\mathbf{x}, \mathbf{y}_p\right)| \leq \sigma_p\|\mathbf{W}_r\left(\mathbf{F}_m - \mathbf{F}\right)\mathbf{x}\|_2$$

The key to bounding this value for an arbitrary $(\mathbf{x}, \mathbf{y})$ is to first upper bound it in terms of the representative points, $\sum_{i=1}^{n}\|\mathbf{W}_r\left(\mathbf{F}_m - \mathbf{F}\right)\mathbf{b}_i\|_2$, and then provide an upper bound on this term with representative points.

**Part 1: Upper bound in terms of representative points**      According to Assumption 5, for all $\mathbf{x}$, w.h.p.

$$\|\mathbf{W}_r\left(\mathbf{F}_m - \mathbf{F}\right)\mathbf{x}\|_2 = \left\|\sum_{j=1}^{n}\alpha_j\mathbf{W}_r\left(\mathbf{F}_m - \mathbf{F}\right)\mathbf{b}_j + \mathbf{W}_r\left(\mathbf{F}_m - \mathbf{F}\right)\eta\right\|_2 \quad (7)$$

Using Cauchy-Schwarz, we further obtain

$$(7) \leq \sqrt{\sum_{j=1}^{n}\alpha_j^2}\sqrt{\sum_{j=1}^{n}\|\mathbf{W}_r\left(\mathbf{F}_m - \mathbf{F}\right)\mathbf{b}_j\|_2^2} + \|\mathbf{W}_r\left(\mathbf{F}_m - \mathbf{F}\right)\|_2\|\eta\|_2$$

$$\leq \sqrt{r}\sqrt{\sum_{j=1}^{n}\|\mathbf{W}_r\left(\mathbf{F}_m - \mathbf{F}\right)\mathbf{b}_j\|_2^2} + \frac{2B_{\mathbf{W}_r}B_{\mathbf{F}}\epsilon}{t} \quad (8)$$

**Part 2: Bounding** $A = \sqrt{\sum_{j=1}^{n}\|\mathbf{W}_r\left(\mathbf{F}_m - \mathbf{F}\right)\mathbf{b}_j\|_2^2}$

Let $B_{\mathrm{L}}(\mathbf{F}_m||\mathbf{F})$ denote the Bregman divergence,

$$B_{\mathrm{L}}(\mathbf{F}_m||\mathbf{F}) = \mathrm{L}\left(\mathbf{F}_m\right) - \mathrm{L}\left(\mathbf{F}\right) - \langle\mathbf{F}_m - \mathbf{F}, \nabla\mathrm{L}\left(\mathbf{F}\right)\rangle \quad (9)$$

where the dot-product notation for the matrices corresponds to element-wise product and summation. We use the following two bounds, proved below.

$$B_N(\mathbf{F}_m||\mathbf{F}) \leq B_{\mathrm{L}}(\mathbf{F}_m||\mathbf{F}) \quad (10)$$
$$B_N(\mathbf{F}||\mathbf{F}_m) \leq B_{\mathrm{L}_m}(\mathbf{F}||\mathbf{F}_m) \quad (11)$$
$$B_{N_b}(\mathbf{F}_m||\mathbf{F}) + B_{N_b}(\mathbf{F}||\mathbf{F}_m) \leq a[B_N(\mathbf{F}_m||\mathbf{F}) + B_N(\mathbf{F}||\mathbf{F}_m)] \quad (12)$$

**Obtaining inequalities in** (10) **and** (11)      The first term comes from the fact that $\mathrm{L} - N$ is strictly convex. This is because the sum of strictly convex functions for $N$ are a strict subset of sum of strictly convex functions for L. Since $\mathrm{L} - N$ is still strictly convex, it provides a valid potential for the Bregman divergence and gives

$$0 \leq B_{\mathrm{L}-N}(\mathbf{F}_m||\mathbf{F}) = B_{\mathrm{L}}(\mathbf{F}_m||\mathbf{F}) - B_N(\mathbf{F}_m||\mathbf{F}) \implies B_{\mathrm{L}}(\mathbf{F}_m||\mathbf{F}) \geq B_N(\mathbf{F}_m||\mathbf{F}).$$

The same reasoning applies to $B_{\mathrm{L}_m}(\mathbf{F}||\mathbf{F}_m) \geq B_N(\mathbf{F}||\mathbf{F}_m)$.

**Obtaining inequality in** (12)      The second term follows from Assumption 6. Notice that the Bregman divergence for $N_b$ and $N$ simplifies as follows.

$$B_{N_b}(\mathbf{F}_m||\mathbf{F}) + B_{N_b}(\mathbf{F}||\mathbf{F}_m)$$
$$= N_b\left(\mathbf{F}_m\right) - N_b\left(\mathbf{F}\right) - \langle\mathbf{F}_m - \mathbf{F}, \nabla N_b\left(\mathbf{F}\right)\rangle + N_b\left(\mathbf{F}\right) - N_b\left(\mathbf{F}_m\right) - \langle\mathbf{F} - \mathbf{F}_m, \nabla N_b\left(\mathbf{F}_m\right)\rangle$$
$$= \langle\mathbf{F}_m - \mathbf{F}, \nabla N_b\left(\mathbf{F}_m\right) - \nabla N_b\left(\mathbf{F}\right)\rangle$$

By the definition of directional derivatives,

$$\langle \mathbf{F}_m - \mathbf{F}, \nabla N_b\left(\mathbf{F}\right)\rangle = \lim_{\alpha \to 0} \frac{N_b(\mathbf{F} + \alpha(\mathbf{F}_m - \mathbf{F})) - N_b(\mathbf{F})}{\alpha} = \lim_{\alpha \to 0} \frac{N_b((1-\alpha)\mathbf{F} + \alpha\mathbf{F}_m) - N_b(\mathbf{F})}{\alpha}$$

and so, because both limits exists,

$$\langle \mathbf{F}_m - \mathbf{F}, \nabla N_b\left(\mathbf{F}_m\right) - \nabla N_b\left(\mathbf{F}\right)\rangle$$
$$= -\langle \mathbf{F}_m - \mathbf{F}, \nabla N_b\left(\mathbf{F}\right)\rangle - \langle \mathbf{F} - \mathbf{F}_m, \nabla N_b\left(\mathbf{F}_m\right)\rangle$$
$$= \lim_{\alpha \to 0^+} \left[ \frac{N_b(\mathbf{F}) - N_b((1-\alpha)\mathbf{F} + \alpha\mathbf{F}_m)}{\alpha} + \frac{N_b(\mathbf{F}_m) - N_b((1-\alpha)\mathbf{F}_m + \alpha\mathbf{F})}{\alpha} \right]$$
$$\leq \lim_{\alpha \to 0^+} a \left[ \frac{N(\mathbf{F}) - N((1-\alpha)\mathbf{F} + \alpha\mathbf{F}_m)}{\alpha} + \frac{N(\mathbf{F}_m) - N((1-\alpha)\mathbf{F}_m + \alpha\mathbf{F})}{\alpha} \right]$$
$$= a\langle \mathbf{F}_m - \mathbf{F}, \nabla N\left(\mathbf{F}_m\right) - \nabla N\left(\mathbf{F}\right)\rangle$$
$$= a\left[ B_N(\mathbf{F}_m||\mathbf{F}) + B_N(\mathbf{F}||\mathbf{F}_m) \right]$$

where the inequality follows from Assumption 6.

**Bounding $A$ using** (10) **-** (12) **and Assumptions 2 and 3**

$$a\left[ B_L(\mathbf{F}_m||\mathbf{F}) + B_{L_m}(\mathbf{F}||\mathbf{F}_m) \right]$$
$$\geq B_{N_b}(\mathbf{F}_m||\mathbf{F}) + B_{N_b}(\mathbf{F}||\mathbf{F}_m)$$
$$= \frac{1}{n} \sum_{i=1}^{n} \left\langle \mathbf{F} - \mathbf{F}_m, \mathbf{W}_r^\top \nabla L_r\left(\mathbf{W}_r\mathbf{F}\mathbf{b}_i, \mathbf{y}_{r,b_i}\right) \mathbf{b}_i^\top \right\rangle - \frac{1}{n} \sum_{i=1}^{n} \left\langle \mathbf{F} - \mathbf{F}_m, \mathbf{W}_r^\top \nabla L_r\left(\mathbf{W}_r\mathbf{F}_m\mathbf{b}_i, \mathbf{y}_{r,b_i}\right) \mathbf{b}_i^\top \right\rangle$$
$$= \frac{1}{n} \sum_{i=1}^{n} \left\langle \mathbf{W}_r\left(\mathbf{F} - \mathbf{F}_m\right)\mathbf{b}_i, \nabla L_r\left(\mathbf{W}_r\mathbf{F}\mathbf{b}_i, \mathbf{y}_{r,b_i}\right) - \nabla L_r\left(\mathbf{W}_r\mathbf{F}_m\mathbf{b}_i, \mathbf{y}_{r,b_i}\right) \right\rangle$$
$$\geq \frac{c}{n} \sum_{i=1}^{n} \|\mathbf{W}_r\left(\mathbf{F}_m - \mathbf{F}\right)\mathbf{b}_i\|_2^2$$

where the inequality comes from the assumption that function $L_r$ is $c$-strongly convex.

Notice now that, because $\nabla L(\mathbf{F}) = \mathbf{0}$ and $\nabla L_m(\mathbf{F}_m) = \mathbf{0}$

$$B_L(\mathbf{F}_m||\mathbf{F}) + B_{L_m}(\mathbf{F}||\mathbf{F}_m)$$
$$= L\left(\mathbf{F}_m\right) - L\left(\mathbf{F}\right) + L_m\left(\mathbf{F}\right) - L_m\left(\mathbf{F}_m\right)$$
$$= \left( L\left(\mathbf{F}_m\right) - L_m\left(\mathbf{F}_m\right) \right) + \left( L_m\left(\mathbf{F}\right) - L\left(\mathbf{F}\right) \right) \qquad (13)$$
$$= \frac{1}{t} \left[ L_p\left(\mathbf{W}_p\mathbf{F}_m\mathbf{x}_m, \mathbf{y}_{p,m}\right) - L_p\left(\mathbf{W}_p\mathbf{F}_m\mathbf{x}_m', \mathbf{y}_{p,m}'\right) \right]$$
$$\qquad + \frac{1}{t} \left[ L_r\left(\mathbf{W}_r\mathbf{F}_m\mathbf{x}_m, \mathbf{y}_{r,m}\right) - L_r\left(\mathbf{W}_r\mathbf{F}_m\mathbf{x}_m', \mathbf{y}_{r,m}'\right) \right]$$
$$+ \frac{1}{t} \left[ -L_p\left(\mathbf{W}_p\mathbf{F}\mathbf{x}_m, \mathbf{y}_{p,m}\right) + L_p\left(\mathbf{W}_p\mathbf{F}\mathbf{x}_m', \mathbf{y}_{p,m}'\right) \right]$$
$$\qquad + \frac{1}{t} \left[ -L_r\left(\mathbf{W}_r\mathbf{F}\mathbf{x}_m, \mathbf{y}_{r,m}\right) + L_r\left(\mathbf{W}_r\mathbf{F}\mathbf{x}_m', \mathbf{y}_{r,m}'\right) \right]$$

Because $L_r$ is $\sigma_r$-admissible by Assumption 2, we have

$$\left| L_r\left(\mathbf{W}_r\mathbf{F}_m\mathbf{x}_m, \mathbf{y}_{r,m}\right) - L_r\left(\mathbf{W}_r\mathbf{F}\mathbf{x}_m, \mathbf{y}_{r,m}\right) \right| \leq \sigma_r \|\mathbf{W}_r\left(\mathbf{F}_m - \mathbf{F}\right)\mathbf{x}_m\|_2.$$

We get a similar result for $L_p$, using Assumption 4, but with $\sigma_p$. Therefore, we can bound (13) above, and get

$$\frac{c}{n} \sum_{i=1}^{n} \|\mathbf{W}_r\left(\mathbf{F}_m - \mathbf{F}\right)\mathbf{b}_i\|_2^2 \leq \frac{a(\sigma_r + \sigma_p)}{t} \left[ \|\mathbf{W}_r\left(\mathbf{F}_m - \mathbf{F}\right)\mathbf{x}_m\|_2 + \|\mathbf{W}_r\left(\mathbf{F}_m - \mathbf{F}\right)\mathbf{x}_m'\|_2 \right] \quad (14)$$

**Putting it all together to get the upper bound on $A$**     From (14), we get

$$\frac{c}{n} A^2 \leq \frac{2a(\sigma_r + \sigma_p)}{t} \left( \sqrt{r}A + \frac{2B_{\mathbf{W}_r}B_{\mathbf{F}}\epsilon}{t} \right)$$
$$\implies A \leq \frac{a(\sigma_r + \sigma_p)n}{ct} \left( \sqrt{r} + \sqrt{r + \frac{4\epsilon c B_{\mathbf{W}_r}B_{\mathbf{F}}}{a(\sigma_r + \sigma_p)n}} \right) \qquad (15)$$

Finally, therefore, again using (8),

$$\begin{aligned}
&|L_p\left(\mathbf{W}_p\mathbf{F}_m\mathbf{x}, \mathbf{y}_p\right) - L_p\left(\mathbf{W}_p\mathbf{F}\mathbf{x}, \mathbf{y}_p\right)| \\
&\leq \sigma_p\|\mathbf{W}_r\left(\mathbf{F}_m - \mathbf{F}\right)\mathbf{x}\|_2 \\
&\leq \sigma_p\sqrt{r}\sqrt{\sum_{j=1}^{n}\|\mathbf{W}_r\left(\mathbf{F}_m - \mathbf{F}\right)\mathbf{b}_j\|_2^2} + \sigma_p\frac{2B_{\mathbf{W}_r}B_{\mathbf{F}}\epsilon}{t} \\
&\leq \frac{a(\sigma_r+\sigma_p)n\sigma_p}{ct}\left(r + \sqrt{r^2 + \frac{4\epsilon c B_{\mathbf{W}_r}B_{\mathbf{F}}r}{a(\sigma_r+\sigma_p)n}}\right) + \frac{2\epsilon\sigma_p B_{\mathbf{W}_r}B_{\mathbf{F}}}{t}
\end{aligned}$$

∎

# B  Examples of specific constants for the Main Theorem

**Corollary 1** *In Assumption 4, if $\mathbf{W}_p \in \mathbb{R}^{m\times k}, \mathbf{W}_r \in \mathbb{R}^{d\times k}, d \geq k \geq m, \mathbf{W}_r$ is full rank, $L_p$ is $\sigma$-admissible, then for $\mathbf{W}_r^{-1}$ the inverse matrix of the first $k$ rows of $\mathbf{W}_r$, $\sigma_p = \sigma\|\mathbf{W}_p\|_F\|\mathbf{W}_r^{-1}\|_F$.*

**Proof:** Since $\mathbf{W}_r$ is full rank, we must have $\mathbf{W}_p = \mathbf{A}\mathbf{W}_r$, where the last $d - k$ columns of $\mathbf{A}$ are all zeros. Hence $\|\mathbf{W}_p(\mathbf{F} - \mathbf{F}_m)\mathbf{x}\|_2 \leq \|\mathbf{A}\|_F\|\mathbf{W}_r(\mathbf{F} - \mathbf{F}_m)\mathbf{x}\|_2$. In the meanwhile, $\|\mathbf{W}_p\mathbf{W}_r^{-1}\|_F = \|\mathbf{A}\mathbf{W}_r\mathbf{W}_r^{-1}\|_F = \|\mathbf{A}\|_F$, where $\mathbf{W}_r^{-1}$ is the inverse matrix of the first $k$ rows of $\mathbf{W}_r$. Hence $\|\mathbf{A}\|_F \leq \|\mathbf{W}_p\|_F\|\mathbf{W}_r^{-1}\|_F$. It implies $\sigma_p = \sigma\|\mathbf{W}_p\|_F\|\mathbf{W}_r^{-1}\|_F$, since

$$\begin{aligned}
&|L_p\left(\mathbf{W}_p\mathbf{F}_m\mathbf{x}, \mathbf{y}_p\right) - L_p\left(\mathbf{W}_p\mathbf{F}\mathbf{x}, \mathbf{y}_p\right)| \\
&\leq \sigma\|\mathbf{W}_p(\mathbf{F}_m - \mathbf{F})\mathbf{x}\|_2 \\
&\leq \sigma\|\mathbf{W}_p\|_F\|\mathbf{W}_r^{-1}\|_F\|\mathbf{W}_r(\mathbf{F} - \mathbf{F}_m)\mathbf{x}\|_2.
\end{aligned}$$

∎

**Corollary 2** *For $L_r$ the least-squares loss,*

$$c = 2 \quad and \quad \sigma_r = 2B_{\mathbf{W}_r}B_{\mathbf{F}}B_{\mathbf{x}} + 2B_{\mathbf{x}}.$$

*If $L_p$ is*

1. *the least-squares loss, then $\sigma = 2B_{\mathbf{W}_p}B_{\mathbf{F}}B_{\mathbf{x}} + 2B_{\mathbf{y}}$*
2. *the cross-entropy, with $\mathbf{y}_p \in \{0, 1\}^m$, then $\sigma = 2\sqrt{m}$*
3. *the cross-entropy, with $\mathbf{y}_p \in \{-1, 1\}^m$, then $\sigma = \sqrt{m}$.*

**Proof:** For the least-squares loss $L_r$, we get $c = 2$ because

$$\begin{aligned}
\langle\mathbf{x}_1 - \mathbf{x}_2, \nabla L_r\left(\mathbf{x}_1, \mathbf{y}\right) - \nabla L_r\left(\mathbf{x}_2, \mathbf{y}\right)\rangle \\
= \langle\mathbf{x}_1 - \mathbf{x}_2, 2\left(\mathbf{x}_1 - \mathbf{x}_2\right)\rangle \\
\geq 2\|\mathbf{x}_1 - \mathbf{x}_2\|_2^2.
\end{aligned}$$

We get $\sigma_r = 2B_{\mathbf{W}_r}B_{\mathbf{F}}B_{\mathbf{x}} + 2B_{\mathbf{x}}$ because

$$\begin{aligned}
&|L_r\left(\mathbf{W}_r\mathbf{F}_1\mathbf{x}, \mathbf{y}_r\right) - L_r\left(\mathbf{W}_r\mathbf{F}_2\mathbf{x}, \mathbf{y}_r\right)| \\
&= \left|\|\mathbf{W}_r\mathbf{F}_1\mathbf{x} - \mathbf{y}_r\|_2^2 - \|\mathbf{W}_r\mathbf{F}_2\mathbf{x} - \mathbf{y}_r\|_2^2\right| \\
&= |\langle\mathbf{W}_r\left(\mathbf{F}_1 - \mathbf{F}_2\right)\mathbf{x}, \mathbf{W}_r\mathbf{F}_1\mathbf{x} + \mathbf{W}_r\mathbf{F}_2\mathbf{x} - 2\mathbf{y}_r\rangle| \\
&\leq \|\mathbf{W}_r\left(\mathbf{F}_1 - \mathbf{F}_2\right)\mathbf{x}\|_2\|\mathbf{W}_r\mathbf{F}_1\mathbf{x} + \mathbf{W}_r\mathbf{F}_2\mathbf{x} - 2\mathbf{y}_r\|_2 \\
&\leq \left(2B_{\mathbf{W}_r}B_{\mathbf{F}}B_{\mathbf{x}} + 2B_{\mathbf{x}}\right)\|\mathbf{W}_r\left(\mathbf{F}_1 - \mathbf{F}_2\right)\mathbf{x}\|_2
\end{aligned}$$

Similarly, for $L_p$ the least-squares loss, $\sigma = 2B_{\mathbf{W}_p}B_{\mathbf{F}}B_{\mathbf{x}} + 2B_{\mathbf{y}}$.

For the case where $L_p$ is the cross-entropy loss, let

$$\begin{aligned}
\mathbf{z} &= \mathbf{W}_p\mathbf{F}_1\mathbf{x} \\
\mathbf{z}' &= \mathbf{W}_p\mathbf{F}_2\mathbf{x} \\
\mathbf{y} &= \mathbf{y}_p
\end{aligned}$$

with $\mathbf{a}_i$ denoting the $i$-th element of vector $\mathbf{a}$. When $\mathbf{y} \in \{0,1\}^m$,

$$|L_p(\mathbf{z}, \mathbf{y}) - L_p(\mathbf{z}', \mathbf{y})|$$

$$= \left| \sum_i^m \left[ -\mathbf{y}_i \ln \frac{1}{1 + \exp^{-\mathbf{z}_i}} - (1 - \mathbf{y}_i) \ln \frac{1}{1 + \exp^{\mathbf{z}_i}} \right. \right.$$

$$\left. \left. + \mathbf{y}_i \ln \frac{1}{1 + \exp^{-\mathbf{z}'_i}} + (1 - \mathbf{y}_i) \ln \frac{1}{1 + \exp^{\mathbf{z}'_i}} \right] \right|$$

$$= \left| \sum_i^m \left[ \mathbf{y}_i \left( \ln \frac{1}{1 + \exp^{\mathbf{z}_i}} + \ln \frac{1}{1 + \exp^{-\mathbf{z}'_i}} \right. \right. \right.$$

$$\left. - \ln \frac{1}{1 + \exp^{-\mathbf{z}_i}} - \ln \frac{1}{1 + \exp^{\mathbf{z}'_i}} \right)$$

$$\left. \left. + \ln \frac{1}{1 + \exp^{\mathbf{z}'_i}} - \ln \frac{1}{1 + \exp^{\mathbf{z}_i}} \right] \right|$$

$$= \left| \sum_i^m \left[ \mathbf{y}_i \ln \frac{\exp^{\mathbf{z}'_i}}{\exp^{\mathbf{z}_i}} + \ln \frac{1}{1 + \exp^{\mathbf{z}'_i}} - \ln \frac{1}{1 + \exp^{\mathbf{z}_i}} \right] \right|$$

$$= \left| \sum_i^m \left[ \mathbf{y}_i(\mathbf{z}'_i - \mathbf{z}_i) + \ln \frac{1 + \exp^{\mathbf{z}_i}}{1 + \exp^{\mathbf{z}'_i}} \right] \right|$$

$$\leq \sum_i^m |\mathbf{y}_i (\mathbf{z}'_i - \mathbf{z}_i)|$$

$$+ \sum_i^m \min \left( \left| \ln \frac{1 + \exp^{\mathbf{z}_i}}{1 + \exp^{\mathbf{z}'_i}} \right|, \left| \ln \frac{1 + \exp^{\mathbf{z}'_i}}{1 + \exp^{\mathbf{z}_i}} \right| \right)$$

$$\leq \|\mathbf{z} - \mathbf{z}'\|_1 + \sum_i^m \min \left( \left| \ln \frac{1 + \exp^{\mathbf{z}_i}}{1 + \exp^{\mathbf{z}'_i}} \right|, \left| \ln \frac{1 + \exp^{\mathbf{z}'_i}}{1 + \exp^{\mathbf{z}_i}} \right| \right)$$

To bound this second component, notice that if $\mathbf{z}'_i \leq \mathbf{z}_i$,

$$\frac{1 + \exp^{\mathbf{z}_i}}{1 + \exp^{\mathbf{z}'_i}} - \frac{\exp^{\mathbf{z}_i}}{\exp^{\mathbf{z}'_i}} = \frac{\exp^{\mathbf{z}'_i} - \exp^{\mathbf{z}_i}}{\exp^{\mathbf{z}'_i}(1 + \exp^{\mathbf{z}'_i})} \leq 0.$$

This implies

$$\left| \ln \frac{1 + \exp^{\mathbf{z}_i}}{1 + \exp^{\mathbf{z}'_i}} \right| = \ln \frac{1 + \exp^{\mathbf{z}_i}}{1 + \exp^{\mathbf{z}'_i}}$$

$$\leq \ln \frac{\exp^{\mathbf{z}_i}}{\exp^{\mathbf{z}'_i}} = \left| \ln \frac{\exp^{\mathbf{z}_i}}{\exp^{\mathbf{z}'_i}} \right| = |\mathbf{z}_i - \mathbf{z}'_i|.$$

Therefore, we get

$$|L_p(\mathbf{z}, \mathbf{y}) - L_p(\mathbf{z}', \mathbf{y})| \leq \|\mathbf{z} - \mathbf{z}'\|_1 + \sum_i^m |\mathbf{z}_i - \mathbf{z}'_i|$$

$$= 2 \|\mathbf{z} - \mathbf{z}'\|_1$$

$$\leq 2\sqrt{m} \|\mathbf{z} - \mathbf{z}'\|_2.$$

For $\mathbf{y}_p \in \{-1,1\}^m$, similarly to the case where $\{0,1\}^m$,

$$\left| \ln \frac{1 + \exp^{\mathbf{y}_i \mathbf{z}'_i}}{1 + \exp^{\mathbf{y}_i \mathbf{z}_i}} \right| \leq |\mathbf{y}_i \mathbf{z}'_i - \mathbf{y}_i \mathbf{z}_i| \leq |\mathbf{z}'_i - \mathbf{z}_i|,$$

then we have

$$|L_p\left(\mathbf{z},\mathbf{y}\right) - L_p\left(\mathbf{z}',\mathbf{y}\right)|$$

$$= \left|\sum_i^m \left[\ln\frac{1}{1+\exp^{\mathbf{y}_i\mathbf{z}_i}} - \ln\frac{1}{1+\exp^{\mathbf{y}_i\mathbf{z}_i'}}\right]\right|$$

$$= \left|\sum_i^m \left[\frac{1+\exp^{\mathbf{y}_i\mathbf{z}_i'}}{1+\exp^{\mathbf{y}_i\mathbf{z}_i}}\right]\right|$$

$$\leq \sum_i^m |(\mathbf{z}_i' - \mathbf{z}_i)|$$

$$\leq \|\mathbf{z} - \mathbf{z}'\|_1$$

$$\leq \sqrt{m}\,\|\mathbf{z} - \mathbf{z}'\|_2$$

∎