[Reviews · NeurIPS 2018]

Reviewer 1



This paper performs an analysis of linear supervised autoencoders, showing that they have nice O(1/t) stability properties that in turn transfers to a nice generalization error bound. They also show that in practice, regularizing a deep model with an autoencoder penalty outperforms L2 and dropout regularizations in some scenarios. This is most notable if the model is allowed more capacity: the capacity is used to fit the reconstruction (with apparent benefits) rather than overfit on the predictive task. This result seems to be an interesting improvement over Liu et al. if one cares about/believes in reconstruction. While I think most in our community would agree that capturing factors of variation through reconstruction is very appealing (and previous literature favorably points that way), there are instances (unfortunately rarely documented) of auxiliary reconstruction losses harming performance or requiring extensive tuning, and the bound that you compute does not necessarily suggest a better performance either. I intuitively agree that reconstruction seems preferable to L2, and as you say, “potentially without strongly biasing the solution”, but I would not be surprised if under certain mild conditions the bias that a reconstruction loss adds was demonstrably larger than that of L2. The empirical part nicely demonstrates the point of the authors (and has a good methodology). Something feels like it is missing though. The main result of this paper is a O(1/t) bound over t the number of dataset samples, yet, there is not a single plot that reflects varying the size of a dataset. The paper itself was well written, and although I had to read a few of the referenced papers to get familiar with stability bounds, the paper felt clear. I think overall this is a neat contribution that could maybe be stronger but is nonetheless relevant to deep learning training understanding. I gave myself a confidence score of 3: while I think I understand the paper and its results, I do not have sufficient understanding of the theory to say with certainty that the math is 100% correct. Minor comments: The equation following (2) has a typo: the first L_p should be a function of x’_m and y’_p,m (rather than x’_i and y’_p,i). wrt Theorem 1. In practice, people rarely bound their parameters (except maybe for RNNs, and even then people usually only clip gradients). Without L2, what stops say B_F or rather ||F|| from growing in O(t)? (Maybe this is wrong since this is for the linear case, but since more data points can require a more complex hypothesis they might require larger weights) CIFAR-10 is a specific subset of 60k examples of a larger dataset (the tiny images dataset), which itself has 80 million examples. I’d either correct the number or explicit what CIFAR-10 is, because I assume that when you “used a random subset of 50,000 images for training and 10,000 images for testing” you only shuffled the actual 60k image dataset, but referencing the 80M images suggests otherwise. It’s great that you do cross-validation. I think mentioning meta-parameter ranges in the appendix would be useful for reproducibility (or the optimal ones you’ve found). I think you could easily do a plot that’s exactly like Figure 3, but with the x axis being the number of samples used, from say 1 to 50k Rebuttal update: I thank the authors for their response. Again I think this could be a great paper and encourage the authors to add varieties of experiments that would make this work even more convincing.

Reviewer 2



The authors analyze the supervised auto-encoder, a model that attempts to output both the target and a reconstruction of the input. They proof uniform stability for linear supervised autoencoders and show, compared to a standard neural network, the performance of the primary task never decreases when introducing the reconstruction loss. The paper starts by proving uniform stability. The authors do this by showing that the shared parameters ("encoder") don't significantly change with the change of one training sample in the training set. This is a nice theoretical result. In the second part of the paper, the authors compare the linear SAE to SAEs that contain a non-linearity (sigmoid, relu) and a version that first transforms the data using radial basis functions. They also include a network that just outputs the target (no reconstruction). The results show that including the reconstruction as a loss helps in all three cases. The authors further show extensive experiments on the CIFAR dataset, using an SAE with a deeper architecture. The authors show that including the reconstruction error again has a regularizing effect. Improving the training accuracy significantly. Under figure 4, the authors mention several results with respect to dropout, l2 regularization and resnet18. I am curious what the exact numbers are and ask the authors to share this in the rebuttal. I am specifically curious about the ResNet18 result (with and without augmentation). Some reported results on CIFAR10 test set are 91% VGG, 92% NIN and 77% Quick-CIFAR, which are all much higher than the results show in figure 4b. In general I think the paper is well written, with extensive experimentation and attention to detail. However I think the empirical results using deep SAE are lacking and I would not agree that the reconstruction error is "such a simple addition" (figure 3), since it nearly doubles the network size and therefore training time. The theoretical result is interesting, however I am curious if there isn't a simpler way to achieve uniform stability for a linear NN. -------------------------- I would like to thank the authors for the extensive feedback and appreciate the additional results. My main concern with the feedback is the following point: "avoided early stopping (as this conflicts with our generalization claims)". However from the paper line 236, there is no theoretical generalization claim for the deep SAE. I think the simplifying assumptions of a simple optimizer and no data augmentation are fair, but it's unclear if the resnet has overfit with no early stopping. Also given the reconstruction is seen as a regularizer, a comparison with other regularizers would have been nice (better than comparing to not regularizing at all). I think the suggestion of showing results for increasing number of data points (by reviewer #1) is strong and it's good to see the authors picked up on it in the feedback. I hold that the experimental section can be much improved, but recognize that the main contribution is the theoretical result. I have therefore increased my score by one point.

Reviewer 3



This paper takes the input reconstruction (autoencoder) as the regularizer to attain the generalisation of neural network models, which is being called supervised auto-encoder (SAE). It first proves an upper bound for showing the uniform stability of linear SAE, hinting good generalisation performance. Then, it shows empirically that the addition of the input reconstruction can consistently make the neural network models learned to generalize well. Strengths: - Given that most of the deep learning work is empirical, this paper shows theoretically that including the input reconstruction error (auto-encoding task) into the objective of linear SAE can theoretically be shown to improve generalisation performance. This contributes to both originality and significance. - For the experiments, when the dimension of the hidden representation is kept increasing, the SAE is shown empirically to be able to achieve persistently good generalisation performance. The experiments also include the use of the different form of non-linear units and the improvement persists. - The paper is well presented and motivated. Specific comments: - Line 127: D is undefined - How is the performance of SAE compared with the use of autoencoder for pre-training?